# KNOWLEDGE TRANSFER THROUGH VALUE FUNCTION FOR COMPOSITIONAL TASKS

## ABSTRACT

Deep Reinforcement Learning methods are sample inefficient when exploring the environment from scratch. In this work, we introduce an approach of knowledge transfer using the value function combined with curriculum learning, which aims to leverage the learning process by transferring knowledge among progressively increasing task complexity. Our main contribution is demonstrating the effectiveness of this approach by modifying the *degrees of freedom* of the target task, breaking it down into simpler sub-tasks, and leveraging learning by transferring the knowledge along the curriculum steps. We empirically demonstrate the broad possibilities of modifying the degrees of freedom of the target task to leverage learning in classical Reinforcement Learning problems and a real-world control task.

## 1 INTRODUCTION

Deep Reinforcement Learning (DRL) (Mnih et al., 2013; 2015), which leverages the scalability of Reinforcement Learning (RL) by combining it with Deep Learning, has demonstrated remarkable success in diverse applications ranging from games (Vinyals et al., 2019; Wurman et al., 2022) to control scenarios (Wei et al., 2017; Yang et al., 2018; Degrave et al., 2022). Nonetheless, DRL encounters challenges in sample efficiency when exploring environments from scratch. These challenges arise from various factors, including the lack of reward signals in environments with sparse and delayed feedback (Bellemare et al., 2016), the expansive exploration space driven by the high-dimensionality of state-action pairs, and the complexity in trading off competing objectives, each may demand a diverse skill set to be accomplished (Mataric, 1994; Hayes et al., 2022).

Organizing and structuring the learning process can significantly enhance sample efficiency, reducing the need for exhaustive exploration across the state space. Curriculum Learning (CL) (Elman, 1993; Bengio et al., 2009; Narvekar et al., 2020) draws inspiration from educational curricula by gradually introducing increasingly complex tasks. Though formulating a curriculum of tasks presents challenges and might demand domain-specific knowledge and control over the environment, it can result in resource efficiency as easier tasks facilitate faster convergence. Additionally, this method can strengthen robustness and enhance generalization by systematically encompassing various data distribution aspects in a meaningful sequence rather than arbitrary sampling (Schaul et al., 2016; Soviany et al., 2021).

In the context of RL, the challenges posed by tasks can be eased by adjusting their *degrees of freedom* (Narvekar et al., 2016). This entails modifying factors like the state-action space, reward structure, episode length, and the initial state distribution. For example, converting a continuous action space into a discrete one can facilitate exploration (Farquhar et al., 2020), and aligning actions with specific objectives can expedite task completion (Wang et al., 2023). In scenarios where positive outcomes are only observed upon reaching the goal, linking intermediate actions to their final result becomes challenging. This issue can be mitigated by changing the initial states of episodes, bringing the agent closer to the target states (Dai et al., 2021), or shaping rewards to observe positive outcomes (Ng et al., 1999; Andrychowicz et al., 2017). As such, a policy proficient at a simpler source task has the potential to harness its acquired knowledge to facilitate the learning of a complex target task.

This work introduces an approach for knowledge transfer across tasks with progressive complexity structured within a curriculum framework. While CL can speed up the learning process by breaking down a target task into simpler subtasks through adjustments in its degree of freedom, transfer

learning leverages the learning by transferring knowledge acquired in an easier task to harness the learning to a target task. Our method employs the value function, which encodes granular information about task execution, and utilizes the softmax function to facilitate the interleaving of actions between source and target policies. By interleaving actions from source and target policies, the latter can play actions from its estimations, an essential mechanism for self-correction Ostrovski et al. (2021). Furthermore, the method allows the expansion of the artificial neural network architecture (ANN) to accommodate an eventual growth in the search space without requiring mappings from different state-action spaces.

Transfer Learning has been an active research topic in RL (Zhu et al., 2020), taking many forms, such as using logged data from an expert policy (Hussein et al., 2017; Levine et al., 2020) to more high-level knowledge, such as partial policies Sutton et al. (1999). Closely related to our work, in (Taylor et al., 2007), the authors proposed *Transfer via inter-task mapping (TVITM)*, which employed the value function to transfer between tasks with different state and action spaces. Our method differs from previous ones by transferring knowledge without reusing artificial neural network (ANN) weights or demanding state/action mappings for different tasks, allowing the search space to grow organically. We empirically evaluate this approach in classical environments sourced from OpenAI Gym (Brockman et al., 2016) and a control task of pump scheduling for water distribution networks (Donâncio et al., 2022). The outcomes highlight the benefits of this strategy in certain domains, yielding policies with better asymptotic performance compared to policies learned from scratch through conventional exploration techniques.

## 2 BACKGROUND

### 2.1 REINFORCEMENT LEARNING

We adopt the framework of episodic Markov Decision Processes (MDPs) (Sutton and Barto, 2018) to model learning tasks, defined as $(\mathcal{S}, \mathcal{A}, \mathcal{P}, r, \gamma, \mathcal{T}, \mu)$ where $\mathcal{S}$ represents the state space, $\mathcal{A}$ the action space, $\mathcal{P} : \mathcal{S} \times \mathcal{A} \to dist(S')$ the transition probability, $r : \mathcal{S} \times \mathcal{A} \to \mathbb{R}$ the reward function, $\gamma \in [0, 1]$ the discount factor, $\mathcal{T}$ the horizon length, and $\mu$ the initial state distribution. The objective of an agent is to learn a policy $\pi(s|a)$ which maps the state space to the action distribution aiming to maximize returns $r(\pi) = \mathbb{E}_{s \sim \mathcal{P}, a \sim \pi} \sum_{t=0}^{\mathcal{T}} \gamma^t r(s_t, a_t)$. This work is independent of the particular value function-based learning algorithm, and for simplicity, we assume Deep Q-Networks (DQN) Mnih et al. (2013), which update the policy $\pi$ represented by the set of weights $\theta$ by minimizing the loss $\delta$:

$$\delta_i(\theta) = \mathbb{E}_{s,a,s',r \sim \mathcal{D}}[r(s,a) + \gamma max_{a'} Q_{\theta^-}(s', a') - Q_\theta(s, a)]^2. \tag{1}$$

In the RL framework, an agent interacts with the environment starting from a state $s_0 \sim \mu$ and at each timestep $t = 0, 1, 2, ..., \mathcal{T}$, takes an action $a \in \mathcal{A}$ given a state $s \in \mathcal{S}$, receiving a reward $r(s, a) \in \mathcal{R}$, and transitioning to a new state $s' \in \mathcal{S}$. DQN achieved remarkable performance by storing these experiences $< s, a, r, s' >$ in the replay memory $\mathcal{D}$ (Lin, 1992) and shuffling them to break their correlation. Furthermore, it also employs a target network with weights $\theta^-$ to estimate future returns. The target network usually updates at frequency $\lambda$ by copying the learning weights $\theta$, a critical component to enhance the stability of the learning process.

### 2.2 DEGREES OF FREEDOM OF A TARGET TASK

CL hierarchically arranges tasks based on complexity levels. In the context of RL, this can be achieved by breaking down a target task into more straightforward source tasks by modifying the *degrees of freedom* of the former (Narvekar et al., 2016). In (Zhu et al., 2020), the authors shed light on various domain differences that can exist between the source task $\mathcal{M}_s$ and the target task $\mathcal{M}_t$.

**The state-space** $\mathcal{S}$: the state-space can be modified by constraining it to filter out irrelevant information, focusing on critical aspects. On the contrary, it can also be augmented, enriching the agent's observation by providing additional information about the environment;

**The action-space** $\mathcal{A}$: the action-space can be constrained to its subspace to reduce the task complexity, for instance, focusing on a meaningful subset of actions for a given subtask;

**The reward function** $\mathcal{R}$: the reward signals can be manipulated to enhance the learning process for some tasks. That is particularly useful in scenarios with sparse and binary rewards. For example, shaping the reward considering the *distance* to some goal can speed up the observation of positive outcomes;

**Transition dynamics** $\mathcal{P}$: the dynamics of the interactions can be affected by, for example, the distinct physical properties between the source task and the target task, thus altering the probability distribution of reaching a state $s'$ by applying a given action $a$ on the state $s$;

**The initial state distribution** $\mu$: source and target tasks can have different initial state distributions. In this way, the agent could have more chances to experience meaningful outcomes early in the learning process;

**The trajectories length** $\mathcal{T}$: the task length can differ between source and target tasks, allowing the agent to interact longer with the environment, quickly reaching states that are rare to observe with the standard task length.

## 3 RELATED WORK

Curriculum Learning (CL) (Elman, 1993; Bengio et al., 2009) systematically organizes the learning process by initially presenting "easier" tasks and gradually increasing the difficulty as the agent's proficiency in these tasks improves. This approach is grounded in the belief that arranging tasks in a meaningful sequence, instead of random sampling, enhances learning effectiveness (Soviany et al., 2021). This spans from manual curriculum design to semi- or fully automated methods. Recent advancements in CL have showcased its efficacy in DRL, improving sample efficiency and enhancing generalization (Portelas et al., 2020). Various strategies can be employed to induce a curriculum, including prioritizing meaningful experiences (Schaul et al., 2016), shaping the reward to provide more reward signals (Ng et al., 1999; Andrychowicz et al., 2017), or exploring unknown states in the environment (Subramanian et al., 2016; Bellemare et al., 2016; Tang et al., 2017; Pathak et al., 2019). Generative Adversarial Networks (GANs) based approaches have recently attained attention due to their capacity to generate curricula automatically. These techniques can entail goal sampling based on the agent's skill level (Sharma et al., 2021), augment reward signals for the learning agent (Campero et al., 2021), or follow a teacher-student goal sampling paradigm (Florensa et al., 2018). Closer to our approach, in (Farquhar et al., 2020; Wang et al., 2023), the authors propose a curriculum to break down a complex task by reducing its action space. That can be achieved, for example, by discretizing a continuous action space, leading to a smaller search space and facilitating exploration.

While CL focuses on structuring and organizing the learning process, knowledge transfer, also known as transfer learning (Taylor and Stone, 2009), aims to leverage knowledge acquired from prior tasks, resulting in resource savings compared to starting from scratch. Transfer learning in RL is an active research area (Zhu et al., 2020), with various forms such as utilizing logged data (Hussein et al., 2017; Hester et al., 2018; Levine et al., 2020), employing a teacher to interleave rollouts with the student policy Dai et al. (2021), using the value function (Taylor et al., 2007), or transferring partial policies (e.g. options) (Sutton et al., 1999; Jinnai et al., 2020). Additionally, agents can have their capabilities enhanced by incrementally expanding the ANN architecture as they tackle new tasks (Rusu et al., 2016b; Yoon et al., 2018), enabling adaptation to novel challenges while retaining previously acquired knowledge (Khetarpal et al., 2022; Abel et al., 2023). Policy distillation-based approaches (Hinton et al., 2014; Rusu et al., 2016a; Teh et al., 2017; Berseth et al., 2018; Czarnecki et al., 2018; Ghosh et al., 2018) can be considered implicit curriculum combined with a transfer learning method that results in a compressed model capable of handling multiple tasks (Guillet et al., 2022). Our work stands apart from previous approaches by modifying the degrees of freedom of the target task and employing the value function for knowledge transfer. In a related context, Self-Imitation via Reduction (SIR) (Li et al., 2021) proposes a compositional approach to solving RL tasks. Their idea involves learning a reduced task and using self-imitation learning to transfer knowledge to the target task.

## 4 METHOD

This work employs the value function to transfer learning between increasingly complex tasks. The process involves modifying the degrees of freedom within a target task to create one or more easier source tasks, which are subsequently sampled based on their complexity. To structure and organize the learning process, we assume the existence of a curriculum generation method that defines these tasks and establishes a sequential hierarchy of complexity for sampling. Once the tasks are defined and their complexity order established, we employ a transfer learning method that relies on the softmax function. This method utilizes the Q-estimations derived from both the source task ($\mathcal{M}_s$) and the target task ($\mathcal{M}_t$) while introducing a temperature parameter $\tau$ to control the entropy between source and target policies when selecting actions. As the curriculum steps unfold, a target task $\mathcal{M}_{t_i}$ becomes the source task $\mathcal{M}_{s_{i+1}}$ for the subsequential step until convergence to a final target task $\mathcal{M}_{t_n}$.

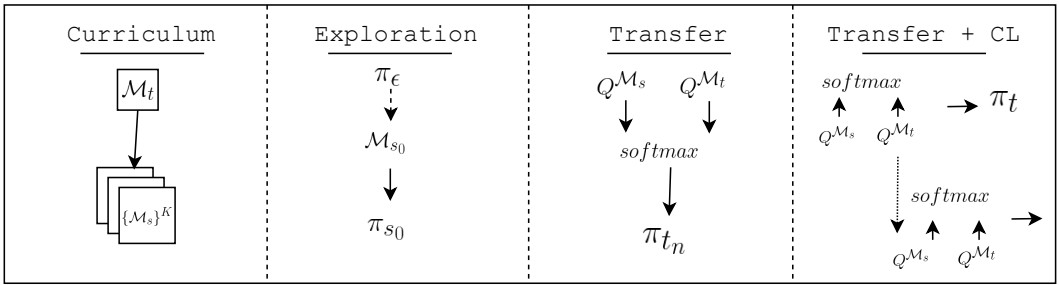

Figure 1: Knowledge transfer steps through a curriculum

### 4.1 KNOWLEDGE TRANSFER METHOD

We introduce two variations of the knowledge transfer methods tailored to the degrees of freedom between tasks $\mathcal{M}_s$ and $\mathcal{M}_t$, specifically regarding their action space $\mathcal{A}$. In the first scenario, we consider unnormalized Q-estimations, denoted as $Q^{\mathcal{M}_s}$ from task $\mathcal{M}_s$ and $Q^{\mathcal{M}_t}$ from task $\mathcal{M}_t$, where both tasks share an identical action space. When selecting an action during the learning process of the target task, we employ the softmax function having as input the maximum output values from both $Q^{\mathcal{M}_s}$ and $Q^{\mathcal{M}_t}$. The application of softmax results in a probability distribution, which guides the selection of the action with the highest value from the respective Q-estimation:

$$\text{softmax}(\{\max(Q^{\mathcal{M}_s}), \max(Q^{\mathcal{M}_t})\}) = \frac{e^{\{max(Q^{\mathcal{M}_s}), max(Q^{\mathcal{M}_t})\}\backslash\tau}}{e^{max(Q^{\mathcal{M}_s})\backslash\tau} + e^{max(Q^{\mathcal{M}_t})\backslash\tau}}$$

$$\text{action } a = \begin{cases} \text{argmax}(Q^{\mathcal{M}_s}), & \text{with probability softmax}_{Q^{\mathcal{M}_s}} \\ \text{argmax}(Q^{\mathcal{M}_t}), & \text{with probability softmax}_{Q^{\mathcal{M}_t}} \end{cases}$$

Figure 1 provides an overview of our knowledge transfer approach. Initially, a target task $\mathcal{M}_t$ into $K$ source subtasks $\mathcal{M}_s$, which are sampled based on complexity. For the sake of simplicity, we focus on one pair of $< \mathcal{M}_s, \mathcal{M}_t >$ at a time, although we can use a set of source policies $\{\pi_{s_i}\}_{i=1}^K$ to leverage the learning to a target policy $\pi_t$. In the first curriculum step, where no source policy is available, we employ an exploration strategy, such as $\epsilon$-greedy, to learn a policy $\pi_{s_0}$. We use this policy $\pi_{s_0}$ to interleave actions with a target policy learning how to perform the subsequent task $\mathcal{M}_{t_0}$[1]. To transfer knowledge between two policies, a *softmax* function is employed, combining Q-estimations from source $max(Q^{\mathcal{M}_s})$ and target $max(Q^{\mathcal{M}_t})$ tasks. Once the state space between the tasks can be different, we need a function $s_{\mathcal{M}_s}(\phi) = h(s_{\mathcal{M}_t}(\phi))$ which contracts or expands the current observations $s_{\mathcal{M}_t}(\phi)$ to match the previously learned state space by $\pi_s$.

---

[1]It's noteworthy that we can use $\mathcal{M}s_1$ and $\mathcal{M}t_0$ interchangeably

In the following, a policy that has mastered a target task $\mathcal{M}_{t_i}$ becomes the source task $\mathcal{M}_{s_{i+1}}$ for the subsequent target task $\mathcal{M}_{t_{i+1}}$, and this process continues until we have unfold all the curriculum steps, ultimately converging to a final target task $\mathcal{M}_{t_n}$. As the complexity of the learning tasks increases, and the search space between two curriculum steps may expand, a necessity arises to augment the artificial neural network (ANN) architecture to accommodate the knowledge requirements. This expansion ensures that the network can effectively absorb and apply the additional knowledge required to master the more intricate tasks.

However, we may have a set of tasks learned simultaneously for a single curriculum step. For that, our second case considers a set of policies $\{\pi_{s_j}\}_{j=1}^K$ transferring knowledge to a target policy $\pi_t$ being the action space $\mathcal{A}(\mathcal{M}_{s_j}) \neq \mathcal{A}(\mathcal{M}_{s_{l \neq j}})$ and $\mathcal{A}(\mathcal{M}_{s_1}) \cup \mathcal{A}(\mathcal{M}_{s_2}) \cup ... \cup \mathcal{A}(\mathcal{M}_{s_K}) = \mathcal{A}(\mathcal{M}_t)$. In other words, the action spaces of two source tasks can be different, but the set of source tasks has the same action spaces as the target task, such that $\mathcal{A}(\mathcal{M}_{s_j})_{j=1}^K = \mathcal{A}(\mathcal{M}_t)$. Thus, Q-estimations for overlapping actions are averaged, while those for distinct actions are combined to obtain $Q^{\mathcal{M}_s}$. Finally, we select actions using the probability distribution outputted by the softmax($\{\max(Q^{\mathcal{M}_s}), \max(Q^{\mathcal{M}_t})\}$).

By employing the value function, which estimates the *quality* of state-action pairs, we can transfer low-level information, which carries out more granular information regarding how to perform a task Taylor et al. (2007). However, the value function can be biased, leading to overestimation when learning passively. By interleaving actions sample from $Q^{\mathcal{M}_s}$ and $Q^{\mathcal{M}_t}$ allows the target task policy to explore its underlying estimations, an essential mechanism for self-correcting (van Hasselt et al., 2016; Ostrovski et al., 2021). Finally, the temperature parameter $\tau$ can lead to distinct entropy between the policies $\pi_{\mathcal{M}_s}$ and $\pi_{\mathcal{M}_t}$ during the transfer learning process. In the following section, we delve into the impact of reward function normalization on this entropy.

## 4.2 THE REWARD NORMALIZATION MATTERS

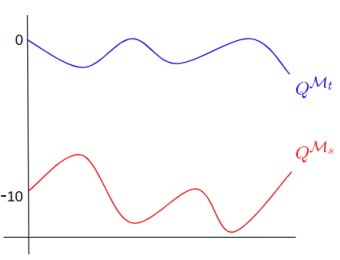

Figure 2: Q-values distribution from source $\mathcal{M}_s$ and target $\mathcal{M}_t$ tasks.

The normalization of the reward function depends on the task. Typically, the scale varies using a normalization $[0, 1]$, where the policy tries to maximize the cumulative rewards, and $[-1, 1]$, where the goal is to minimize negative outcomes and maximize positive ones. In some tasks, normalizing the reward using negative values can be valuable to facilitate interpretation by the learning algorithm, highlighting penalized actions. However, if most of the rewards observed during an episode's rollout are negatives, the Q-estimations for this task would also inherently output negative values.

Since our approach relies on the magnitude of the value function to sample actions, the normalization scale directly impacts the entropy between source and target policy. As illustrated in Figure 2, a source policy that mostly observes negative rewards, even though it performs the task well, has lower Q-estimations than a target policy started with random weights without any training. Thus, most actions in the early training come from this suboptimal target policy, hampering the overall transfer learning performance. To mitigate this problem, we propose to combine estimates $Q^{\mathcal{M}_s}$ and $Q^{\mathcal{M}_t}$ when updating estimates of the former while learning a target task by modifying the Equation 1:

$$\delta_i = \mathbb{E}_{s,a,s',r \sim \mathcal{D}}[r(s,a) + \gamma max_{a'}(\alpha Q^{\mathcal{M}_t}(s',a') + \beta Q^{\mathcal{M}_s}(s',a')) - Q^{\mathcal{M}_t}(s,a)]^2.$$

Thus, when predicting Q-values for the future state, the estimates from the source and target tasks are combined such that $Q(s',a') = \alpha Q^{\mathcal{M}_t}(s',a') + \beta Q^{\mathcal{M}_s}(s',a')$, being $\alpha + \beta = 1$ and $\alpha > \beta = 1$. This linear combination leverages the knowledge transfer process since the $Q^{\mathcal{M}_t}$ estimations have a dependency with values $Q^{\mathcal{M}_s}$, which can be seen as a form of reward shaping.

## 5 EXPERIMENTS

### 5.1 ENVIRONMENTS

To demonstrate the effectiveness of our approach, we conducted experiments on classical reinforcement learning (RL) problems sourced from the OpenAI Gym library (Brockman et al., 2016) as well as a real-world control task centered on optimizing pump scheduling within a water distribution system (Donâncio et al., 2022). In the following, we outline the specific adjustments made to the degrees of freedom for each task, with further comprehensive details about the environments available in section A:

**Mountain Car:** to facilitate the learning of the source task, we adjust the degree of freedom regarding the transition probability $\mathcal{P}$ by manipulating the task's physical properties concerning friction. This modification enables the agent to achieve higher speeds rapidly. Additionally, we extended the episode length $\mathcal{T}$ from 200 to 5000 timesteps to facilitate exploration. During the exploration, the probability $\epsilon$ of taking random actions is gradually reduced from 1 to 0.01, decaying every time the goal is reached.

**Taxi Cab:** in our approach to the Taxi Cab environment, we introduce a reward shaping mechanism that rewards the agent with +10 for passengers' successful pickup and drop-off during the source task. Furthermore, we introduce a bonus to the reward at each timestep, considering the grid distance required to achieve these subgoals. The exploration parameter $\epsilon$ gradually decreases using a decay factor considering the number of episodes, initially set at 1 and eventually reaching 0.2.

**Frozen Lake:** In the Frozen Lake scenario, our approach begins by initiating the source task with a modified initial state that situates the agent closer to the goal state. Also, mirroring the Mountain Car strategy, we implement an $\epsilon$-greedy decay schedule that reduces the exploration rate $\epsilon$ each time the agent successfully reaches the goal. As $\epsilon$ decreases and eventually falls below a predefined threshold, we transition back to utilizing the standard initial state while continuing to learn the source task.

**Pump Scheduling:** The reward function proposed for pump scheduling encapsulates three subgoals: (i) diminishing electricity consumption of pump operation; (ii) the maintenance of safe water tank levels; and (iii) discouraging frequent pump switches (ON/OFF) while distributing their usage. The task involves managing four distinct pumps with varying supply capacities and electricity consumption profiles. Moreover, a tank with a 10m length is used for water storage. To tackle this problem, we devise a curriculum-based approach divided into three sequential steps. Initially, we train individual policies, focusing on controlling a single pump with a reduced state space, limited to observations related to tank levels. Subsequently, we transfer the acquired knowledge to the subsequent task, which introduces observations related to pump operation, including running time and switches. The final step entails combining these distinct individual policies into a comprehensive one capable of managing all four pumps while incorporating observations about their operation.

Our objective in proposing these curricula is to establish a proof-of-concept that focuses not on identifying the one with the highest learning enhancement but rather on demonstrating the efficacy of transfer learning by modifying various aspects of a target task's degrees of freedom.

### 5.2 RESULTS

In Figure 3, we present the results obtained from experiments conducted using OpenAI's scenarios. The depicted outcomes illustrate the average, maximum, and minimum returns across 10 runs for policies trained on the source task ($\pi_0$) with modified degrees of freedom, policies tackling the target task from scratch ($\pi_\epsilon$) employing $\epsilon$-greedy exploration, and policies addressing the target task via transfer learning ($\pi_1$) while varying the temperature parameter $\tau$. Notably, the MDP employed throughout each curriculum step for learning and evaluation remains largely consistent, except for the reward function, which aligns with those of the target task to establish a grounded comparison.

Our outcomes when learning from scratch in the Mountain Car scenario align with findings reported in (Farquhar et al., 2020). Specifically, the use of $\epsilon$-greedy exploration in the Mountain Car task often struggles to achieve the goal due to the limited exposure to positive outcomes. However, our curriculum-based approach significantly enhances the learning process for the source task, as evident in Figure 3(a). Subsequently, the transfer learning process retrieves that performance being the

discrepancy in cumulative returns due to different physical settings between the source and target task.

As depicted in Figure 3(b), our curriculum approach applied to the Taxi Cab problem demonstrates limited learning enhancement for the source task. We attributed this to the intrinsic exploration challenges posed by the task itself. Consequently, when transitioning from this suboptimal policy obtained in the source task, the transfer learning method fails to surpass the performance achieved through exploration from scratch. It's worth noting that we don't view this outcome as a limitation of our approach but rather as an indicator of the importance of well-designed source tasks to expedite the learning process.

The results for the Frozen Lake scenario are depicted in Figure 3(c), showing an "dip" in performance in the range between 750~1250 episodes resulting from the modification in the initial state during the source task learning. However, after a few subsequent episodes, the agent could once again attain the goal, starting from a state identical to that of the target task. Remarkably, our knowledge transfer approach once more effectively leverages the target task's learning process, resulting in superior performance compared to the target task learned from scratch utilizing $\epsilon$-greedy exploration.

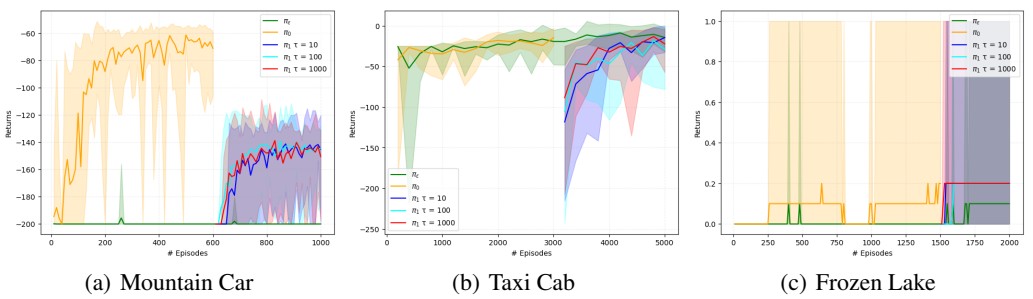

| (a) Mountain Car | (b) Taxi Cab | (c) Frozen Lake |

Figure 3: OpenAI Gym results

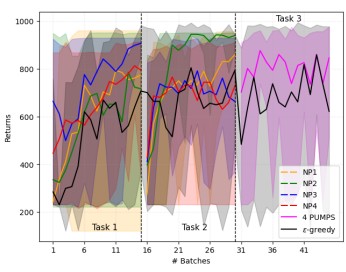

Figure 4: Pump scheduling results

In Figure 4, we compare the results for the pump scheduling problem with policies learned using $\epsilon$-greedy exploration with an ANN architecture identical to the curriculum's final step. We split the three-year dataset regarding water consumption into two years for the learning process and one for policy evaluation. This water consumption data serves as input at each timestep for the water distribution simulator once other states' features are fully transition-dependent. To address fairness regarding samples, we divide the training in "batches" once our curriculum approach learns four individual policies during Task 1 ($\mathcal{M}_1$) and Task 2 ($\mathcal{M}_2$). As such, each batch of policies with action space $\mathcal{A} = \{\text{NOP}, \text{NP\#}\}$ has a single episode, while for $\epsilon$-greedy and Task 3 ($\mathcal{M}_3$) contains four episodes. We employ a reset mechanism for each episode rollout, having the initial tank levels defined by the logged data, matching the water consumption at $t_0$. Finally, $\epsilon$ decays from $1 \rightarrow 0.1$ with a discount factor of 0.95 for $\mathcal{M}_1$ and 0.85 for $\epsilon$-greedy exploration.

The results display the average, maximum, and minimum returns for policy performance after updates using each batch. We have observed that policies that control single pumps (Task 1 and 2) demonstrate better jumpstart performance and lower variance than the full action space learned through $\epsilon$-greedy. Additionally, although a full-action space (Task 3) policy achieved a lower average return than some single-pump policies, which could be explained by the reduced search space, it holds the highest peak for maximum return. Finally, for Task 3, our approach attained a maximum average return peak with lower variance than a policy learned from scratch through exploration.

### 5.3 Ablation Analyses of Q-weighted Combination

We performed an ablation study to assess the effectiveness of the Q-weighted sum in composing $Q(s', a')$. We employ a single source policy $\pi_0^*$ that exhibited the highest cumulative reward over the learning process of $\mathcal{M}_s$ to transfer learning to target policies, one employing the Q-weighted sum (red line) and the other without (blue line). Furthermore, we set fixed values for $\alpha = 0.7$ and $\beta = 0.3$ to assess the method's robustness across all scenarios. Lastly, the black dashed line ($\pi_1$) works as a baseline representing the maximum average peak observed in the previous experiment employing the same temperature value $\tau$, transferring using any source policy $\pi_0$.

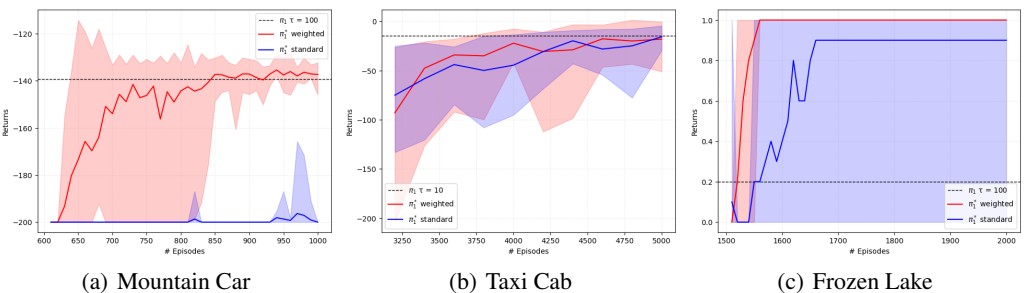

| (a) Mountain Car | (b) Taxi Cab | (c) Frozen Lake |
|---|---|---|

Figure 5: Ablation analyses of Q-weighted sum.

The results in Figure 5 show that the Q-weighted sum can be the cornerstone to effective knowledge transfer in scenarios such as Mountain Car, where most observed rewards are negative. Indeed, our transfer learning strategy can work in two manifolds: interleaving actions from source and target policies while shaping the reward by transferring Q-estimations. In addition, the source policy's performance can substantially impact leveraging the learning process. As seen in the results for the Frozen Lake (see Figure 5(c), our approach could retrieve the optimality of the source policy $\pi_0^*$ which its final step has the same settings as the target task, in very few episodes.

## 6 Conclusions

This work introduces a knowledge transfer approach to effectively leverage the learning process for solving challenging reinforcement learning tasks. The fundamental idea revolves around knowledge transfer through a curriculum of tasks that systematically increase in complexity. We assume a curricula generator method responsible for reducing target task complexity by manipulating its degrees of freedom and hierarchically organizing the derivated simpler sub-task(s). As the curriculum steps unfold, we incorporate the softmax function with a parameter $\tau$, which employs the value function for knowledge transfer across these tasks. Given that our approach relies on the magnitude of value function estimations for action selection, we also present a solution to address the challenge of reward range normalization. We conduct extensive evaluations across various classical reinforcement learning environments and control tasks to empirically validate our approach. These experiments serve as a proof-of-concept by modifying the degrees of freedom in different target tasks, and our results provide compelling evidence of the effectiveness of our approach across four distinct environments.

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

# APPENDIX

## A    FURTHER ENVIRONMENTS DESCRIPTIONS

In this section, we provide a comprehensive overview of each stage within the curricula that were designed to assess the effectiveness of our knowledge transfer approach.

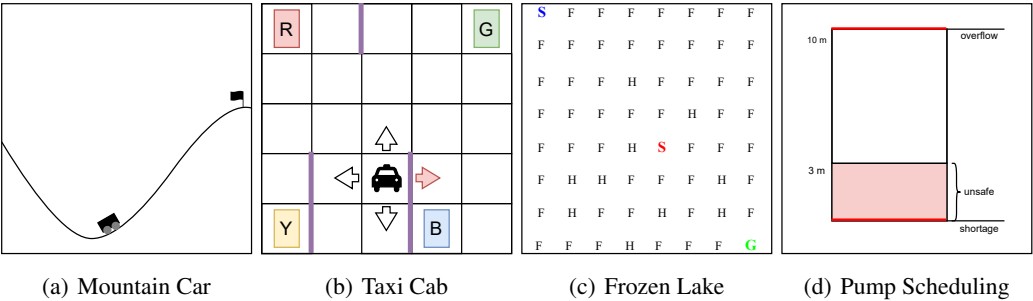

(a) Mountain Car        (b) Taxi Cab        (c) Frozen Lake        (d) Pump Scheduling

Figure 6: Environments.

### A.1    MOUNTAIN CAR

The Mountain Car scenario Moore (1990) presents a challenging task where the agent must navigate a valley to reach the "top of the right hill" (see Figure 6(a)). With discrete actions for acceleration (right, none, left), the agent faces limited positive reward observations and the need to balance accelerations to gain momentum toward reaching the goal. Each episode typically lasts for 200 timesteps.

In this scenario, the absence of positive rewards leads to challenges in achieving optimal performance, and their lack during early exploration phases may result in convergence to suboptimal actions. To address this, we propose breaking down the task through a curriculum. Initially, the agent learns the task with a modified friction force, enabling faster speed gain. Moreover, we extend the episode duration by removing the reset mechanism until policy updates, allowing continuous interaction with the environment for up to 5000 timesteps. Using this strategy, we aim to leverage knowledge acquired in the modified setting to improve performance when transitioning to the standard task with original physics and episode length.

### A.2    TAXI CAB

The Taxi Cab scenario (Dieterich, 2000) involves picking up and dropping off passengers at specified locations R, Y, G, B in Figure 6(b). The agent navigates a grid environment with actions (north, south, west, east, pickup, dropoff) while avoiding internal walls (purple lines). For each illegal pickup or dropoff action, the agent receives a negative reward of -10; otherwise, a positive reward of +20 if the passenger is delivered to the correct location. The agent also gets a -1 penalty for any state-action pair that does not overlap previous rewards. During the exploration, we allow the agent to perform up to 100 actions while evaluating up to 25.

A significant learning challenge arises due to the delayed positive rewards, which only occur after successful dropoffs. Consequently, the agent may struggle to grasp the importance of passenger pickups. To address this issue, we propose a curriculum where the source task uses a modified reward function, receiving a positive reward of +10 for picking up a passenger and an additional +10 for successfully dropping off the passenger at the correct location. Furthermore, to guide the agent efficiently, we introduce a reward bonus based on the distance $d$ to the current goal (pickup or dropoff). For cardinal direction actions (north, south, west, east), the reward is determined by $r = -1 + 1/(d + 1)$. This modification allows the agent to recognize the significance of passenger pickups as part of the task, while the reward bonus based on distance aids navigation toward each subtask. Subsequently, we transfer the learned policy with modified rewards to a policy learning task using the standard reward function.

### A.3 FROZEN LAKE

The Frozen Lake scenario involves guiding the agent from the starting point (S) in the top left corner of a grid to the ending point (G) on the right bottom corner, avoiding falling in holes (H). Each movement action (north, south, west, east) yields a 0 reward while reaching the objective provides a +1 reward. The challenge lies in the absence of positive reward, necessitating the agent to explore and discover a sequence of actions that lead to successful goal-reaching while avoiding premature episode endings caused by falling into holes.

We introduce a curriculum where we initially learn a source task by penalizing premature episode endings through a negative reward of -1 for landing in a hole (H). Additionally, we implement a modification to the initial state, relocating the agent from the top left corner (S) to a more centrally situated state (S), as illustrated in Figure 6(c). As the agent observes positive rewards upon reaching the goal, the likelihood $\epsilon$ for random actions diminishes over time. We use the standard initial state once the $\epsilon$ value becomes $\leq 0.36$ while still learning the source task. Subsequently, we transfer the acquired knowledge to a target task featuring the standard initial state configuration. By manipulating the initial state, we aim to bring the agent closer to the goal state, reducing the need for exploration by facilitating the observation of positive rewards.

### A.4 PUMP SCHEDULING

In (Donâncio et al., 2022), the authors introduce the pump scheduling scenario and its representation as a Partially Observable MDP (POMDP). This control optimization problem relies on minimizing the energy consumption of a set of pumps while meeting the safety constraints of water reservoirs. In this scenario, four pumps (NP1 to NP4) pump water into the system. These pumps have different sizes and electricity consumption. Moreover, a tank with a 10m length is used for water storage with constraints depicted in Figure 6(d). The proposed reward function for the pump scheduling scenario is given by:

$$r_t = -e^{(-1/kW_t)} - B * \psi + \log(1/(P + \omega)),$$

where the term $-e^{(-1/kW_t)}$ penalizes the electricity consumption ($kW$) associated with the operation of pumps $\{(\text{NP\#})\}_{\#=1}^4$ at timestep $t$. Secondly, $B * \psi$ penalizes the agent when the tank levels fail to meet safety constraints. Specifically, safety operation necessitates maintaining the tank's water level at a minimum of 3m. The term $\log(1/(P + \omega))$ aims to lead to an action distribution and avoid pump switches once it decreases the asset's lifetime. Here, $P$ is a non-markovian feature representing the cumulative time the current action has been applied to the system throughout the episode. Lastly, the variable $\omega$ is a penalty for action switches, encouraging the maintenance of more continuous pump operation.

To trade off these sub-goals, the agent needs to obtain information about the environment through states/observations $s \in S$ defined by the following set of features $\phi$, where:

- $T$ is the tank level at some timestep $t$;
- $O$ is the sensor's data regarding water consumption for some timestep $t$;
- $\Gamma$ is time of day;
- $X$ is the last action performed;
- $P$ is the cumulative time that the current action has been performed along the episode;

Once an observation is triggered, the agent has to decide between choosing (or keeping) one of the pumps running or turning (or keeping) it off. At each timestep, only one of the pumps can be running, or none of them (NOP). Thus, the action space is such that $\mathcal{A} = \{\text{NP1, NP2, NP3, NP4, NOP}\}$.

Our approach to addressing the pump scheduling problem follows an incremental strategy, introducing sub-goals one by one, progressively expanding the source task to encompass the target task. To achieve this, we first establish a task where a set of policies is individually trained to control a single pump while meeting the tank level constraints. In this initial task $\mathcal{M}_1$, the observation space is defined as $S = \phi(T, O)$, while the action space consists of $\mathcal{A} = \{\text{NOP, NP\#}\}$.

Once the task $\mathcal{M}_1$ is *mastered*, we transfer the knowledge to the target task $\mathcal{M}_2$, where observations extend to the full state representation $s_t = \phi(T_t, O_t, \Gamma_t, X_t, P_t)$. The next step involves combining these source tasks from $\mathcal{M}_2$ into a final target task. To achieve this, we average the Q-values for each NOP action once each policy maps its estimation. We then combine the estimations for each action using different pumps, resulting in the estimation $Q^{\mathcal{M}_s}$:

$$Q^{\mathcal{M}_s} = \{avg(Q(s, \text{NOP})), Q(s, \text{NP1}), Q(s, \text{NP2}), Q(s, \text{NP3}), Q(s, \text{NP4})\},$$

Finally, the softmax using $Q^{\mathcal{M}_s}$ and $Q^{\mathcal{M}_t}$ is used to output actions while learning the last curriculum step $\mathcal{M}_3$. This enables the agent to effectively leverage the knowledge acquired from the policies controlling individual pumps to address the complete pump scheduling problem. Along with these curriculum steps, the network architecture is adapted to accommodate the increase in task complexity, as shown in the implementation and training details B.

## B   IMPLEMENTATION AND TRAINING DETAILS

### B.1   LEARNING ALGORITHM

We employ Double DQN (DDQN) (van Hasselt et al., 2016) as a learning algorithm in this work. DDQN mitigates the overestimation problem for under-represented state-action pairs by selecting the action using the current Q-estimations but updates it using the target net $\theta^-$ values:

$$\delta_i(\theta) = \mathbb{E}_{s,a,s',r \sim \mathcal{D}} \left[ r(s,a) + \gamma Q_{\theta^-} \left( s', \arg\max_{a'} Q_\theta(s', a') \right) - Q_\theta(s, a) \right]^2. \tag{2}$$

Thus, we employ the linear combination $Q_{\theta^-} = \alpha Q^{\mathcal{M}_t}_{\theta^-} + \beta Q^{\mathcal{M}_s}_\theta$ using learning weights $\theta^-$ from the task $\mathcal{M}_t$ and consolidate ones $\theta$ from the task $\mathcal{M}_s$, to update the estimations $Q^{\mathcal{M}_t}$.

### B.2   ANN ARCHITECURE

**Mountain Car:** The neural network architecture for this task consists of dense layers (64, 64) with ReLU activation functions. The training utilizes an Adam optimizer with a learning rate $\alpha$ of 0.001, a minibatch size of 64, target network updates every 100 samples and a discount factor $\gamma$ of 0.99.

**Taxi Cab:** For the Taxi Cab problem, the neural network architecture includes an embedding layer followed by dense layers (50, 50, 50) using ReLU activation functions. Training employs an Adam optimizer with a learning rate $\alpha$ of 0.001, a minibatch size of 32, target network updates every 500 samples, and a discount factor $\gamma$ of 0.99.

**Frozen Lake:** In the case of Frozen Lake, the neural network architecture features an embedding layer followed by dense layers (128, 64) with ReLU activation functions. The training process uses an Adam optimizer with a learning rate $\alpha$ of 0.001, a minibatch size of 64, target network updates every 100 samples and a discount factor $\gamma$ of 0.99.

**Pump Scheduling:** The ANN architecture in the pump scheduling task varies in size across different curriculum steps. The architecture consists of layers (LSTM, Dense, Dense), and all dense layers have the ReLU activation function. The same architecture is employed for the policy learned using $\epsilon$-greedy from scratch and the task $\mathcal{M}_3$. The training uses an Adam optimizer with a learning rate $\alpha$ of 0.00025, a minibatch size of 32, target network updates every 300 samples, $L_2$ regularization (dense layers) of 0.000001, $\epsilon$-greedy policy interpolation ranging from 1 to 0.1, and a discount factor $\gamma$ of 0.99. The expansion of the ANN architecture's hidden nodes for each task is illustrated in Table 1.

| Actions $\mathcal{A}$ | Task $\mathcal{M}_1$ | Task $\mathcal{M}_2$ | Task $\mathcal{M}_3$ |
|---|---|---|---|
| (NOP, NP1) | (25, 25, 25) | (75, 75, 75) | – |
| (NOP, NP2) | (25, 25, 25) | (75, 75, 75) | – |
| (NOP, NP3) | (50, 50, 50) | (75, 75, 75) | – |
| (NOP, NP4) | (75, 75, 75) | (75, 75, 75) | – |
| Full action space | – | – | (125, 125, 125) |

Table 1: ANN architecture progress for the pump scheduling problem

