# OpenReview forum: "Knowledge Transfer through Value Function for Compositional Tasks"
_ICLR.cc/2024/Conference — Submitted to ICLR 2024_

### Official Review · Reviewer_QcHM · 2023-10-30

**Soundness:** 2 fair
**Presentation:** 2 fair
**Contribution:** 2 fair
**Rating:** 3
**Confidence:** 4

**Summary:**

This paper introduces a novel knowledge transfer method that merges value functions with course learning. To assess the efficacy of this approach, extensive evaluations are conducted across four distinct environments. While the paper offers a thorough and comprehensive explanation of the methodology, the overarching contribution of the article remains somewhat obscure. Furthermore, several sections of the paper appear hastily written, rendering them challenging to comprehend.

**Strengths:**

I believe the paper could benefit from a more comprehensive exploration of related works, as well as a clearer summary of its contributions. Additionally, it would be valuable to provide richer research context to enhance the overall quality of the paper. The advantages and disadvantages of this article are summarized as follows:

Pros:

- The methodology section is introduced thoroughly and is presented in a clear and easily understandable manner.
- The article assesses the effectiveness of the proposed method through extensive empirical evaluations across a range of classic reinforcement learning environments and control tasks.

**Weaknesses:**

Cons：

- Placing the statement 'Our main contribution is demonstrating the effectiveness...' in the abstract section creates confusion regarding the primary contribution of the article, which could be expressed more explicitly.
- The comparison between this article and existing work can be analyzed in detail in the introduction section, which helps readers understand the contribution of this article.
- The paper lacks sufficient information on implementation details, such as the generation of source tasks and hyperparameter settings.
- Too few ablation experiments and comparative experiments have made me doubt the scalability of the method, such as the difficulty of adjusting $\alpha$ and $\beta$ between different tasks.

**Questions:**

For "Transfer+CL" in Figure 1, where the second $ Q ^ {Ms} $should come from $ pi_ What about t $? Is the objective strategy Q function that converged on the previous course step used as the source strategy Q function for the next course step?

---

### Official Review · Reviewer_xTEq · 2023-10-30

**Soundness:** 2 fair
**Presentation:** 1 poor
**Contribution:** 2 fair
**Rating:** 3
**Confidence:** 3

**Summary:**

The paper proposes to use a policy transfer strategy to gradually improve the performance towards a target task under a curriculum learning paradigm. The strategy mixes the estimate of Q functions of the source (or the target of previous iteration) and the target task setups. The setups may differ with an altering task complexities, which are reflected in the state and action space, transition model and reward sparsity. Concretely, the mixing strategy adopts a softmax operation which is adjusted by a temperature parameter. The idea is validated in learning a set of simple gym environments, such as mountain car and taxi cab. Part of the results support a favoured learning efficiency with the introduced curriculum transfer strategy.

**Strengths:**

1. The paper appears to target a general strategy that may handle different categories of task complexity shift.
2. The idea of mixing Q function estimate as the form of knowledge reuse looks reasonable.

**Weaknesses:**

1. It would be very helpful to improve the writing to make it clear what is the technical contribution.
2. It is hard to tell what are the formal and quantitative definition of the task complexities. The paper seems particularly ambitious to cover as many variations as possible while somehow looses the focus. The validation seems solely focusing on some of the task variations while not demonstrating on transferring between different state/action spaces.
3. The validation assumes the existence of a strategy to generate a curriculum, which is actually central to curriculum learning. If the investigation is about what transfer strategy to use, it would make more sense to benchmark other competing transfer strategies. The baselines like \pi_1 look weak.
4. The results are hard to interpret their significance. That is probably due to visualisation, e.g. in Fig. 4, showing extremely large variance. It is hard to tell the competing methods apart due to the overlap. It might be more readable to use table to report statistics at a selected set of episode steps.
5. The benchmarking tasks could be stronger. Curriculum learning has demonstrated in learning with more complex environment dynamics. Demonstrating efficacy could significantly improve the paper's soundness and significance.

**Questions:**

1. Could the paper provide more theoretical insights on the design of transfer strategy so as to be more informative on its advantage comparing to other potential transfer options? The way of mixing multiple Q-networks appear to resemble Duelling DQN which is the base RL algorithm used in the paper. Is there any particular association to this design choice? What might be the correlation to the theoretical motivation of Duelling DQN?
2. Is there a more unified and formal way of presenting the task degree-of-freedom. What types of scope we can expect this strategy might be able to handle beyond the examples presented here?
3. The validation tasks look similar in terms of the problem scale and underlying task goal. Could the paper provide more evidence on problems with a larger scale and different task essences?

---

### Official Review · Reviewer_Jyv8 · 2023-10-31

**Soundness:** 3 good
**Presentation:** 3 good
**Contribution:** 2 fair
**Rating:** 5
**Confidence:** 4

**Summary:**

The authors propose a new method for knowledge transfer across tasks. The new method differs from previous approaches in the way it modifies the degrees of freedom of the target task and employs the value function for knowledge transfer. The method leverages curriculum learning to breakdown the complex tasks to simpler subtasks based on its degree of freedom. Specifically, the process involves modifying the degrees of freedom within a target task to create one or more easier source tasks. The value function encodes granular information about task execution and a transfer learning method is used to learn the complex tasks from easier tasks. The actions between source and target policies are interleaved for self-correction. One key difference from previous methods is that the authors do not reuse artificial neural network (ANN) weights or demanding state/action mappings for different tasks. Empirical results are provided using classical reinforcement learning (RL) problems sourced from the OpenAI Gym library by making adjustments to the degrees of freedom for each task. For example, for the mountain car problem, the authors adjust the degree of freedom for the transition function related to the friction. The authors also include ablation analysis of q-weighted combination.

**Strengths:**

The originality of the paper stems from combining previously known methods in a novel way using degrees of freedom. The paper is well-written. The terms are clear, and well-described. The related work section provides adequate ground work on how the authors work leverages prior work and differs from it. The authors address practical concerns such as self-correction and reward normalization for various sub-tasks.

**Weaknesses:**

The main contribution of the paper is to adjust the degrees of freedom for knowledge transfer yet the empirical results are focused on the proof-of-concept rather than varying degree of freedom, and how that impacts the algorithm performance. There are no theoretical proofs to strengthen the paper in this regard. It feels like the paper is incomplete without supporting evidence whether empirical or theoretical.

**Questions:**

In general, I feel like the proof-of-concept setup is well thought out but incomplete. It would be good to include examples when one or more degrees of freedom is modified. Maybe more complex examples are needed to convey the contribution empirically in the absence theoretical convergence proofs.

---

### Official Review · Reviewer_mFvk · 2023-11-01

**Soundness:** 1 poor
**Presentation:** 1 poor
**Contribution:** 1 poor
**Rating:** 3
**Confidence:** 4

**Summary:**

This paper proposes an approach for curriculum learning in reinforcement learning in which the action-value function is used to progressively transfer knowledge between subsequent tasks. Actions from policies trained in the source and target tasks are blended together according to their normalized action-values. The proposed approach is evaluated over several standard RL benchmark environments as well as a real-world pump scheduling task.

**Strengths:**

* Curriculum learning is an important aspect of reinforcement learning, given the non-stationarity of the policy/data distribution used for training. The general direction of improving curriculum learning is valuable, although I think this paper would benefit from more clearly identifying what specific problem within curriculum learning is being addressed.

**Weaknesses:**

* The motivation and the contributions of this work are not clear to me. While curriculum learning is certainly an important topic within reinforcement learning, I do not understand the approach being proposed in this paper. On top of the many technical concerns I have (see below), I don't understand why such a knowledge transfer mechanism is needed. If the state-action space is the same between tasks, why not simply train each task until convergence (or some other training heuristic) and then move on to training the subsequent task in the curriculum as is commonly done? If the state-action space is not the same, then this method will also need to leverage some method to deal with it just as standard curriculum learning would (e.g. through bootstrapping, action advising, policy distillation, etc).
* The technical details of the proposed approach are also not clear to me. Early in training in the target task, the target Q-function will not have converged and will have poor estimates. And if we use the Q-function from the source task, wouldn't it be out-of-distribution in the target task? In addition, why would we assume that the Q-value in the source task is a proxy for a good action in the target task, when the tasks differ? Under what conditions is this expected to work? Why do this sort of blending at all, why not just start with the policy from the source task and perform standard RL training in the target task?
* The experimental results do not include comparisons to any sort of baseline, including a standard curriculum training setup where policies are simply trained to convergence in a sequence of tasks. The figures are also not clear to me (e.g. Figure 3), I am not sure what I am supposed to take away from them and they are quite difficult to read. For example, the text says the learning process is significantly enhanced but what indicates this?

**Questions:**

1) Why would we need this sort of knowledge transfer mechanism? Why not simply sequentially train a policy to convergence on each task in the curriculum (maintaining the weights between each task)?
2) If we do desire this knowledge transfer, would the Q-function estimates be accurate? If we are learning the Q-function in each task, we wouldn't have an accurate estimate until well into training (at which point, would we still need to transfer knowledge?), so wouldn't these be inaccurate estimates? Won't the source task Q-function be out-of-distribution in the target task as well?

---

### Official Review · Reviewer_Xw5P · 2023-11-04

**Soundness:** 2 fair
**Presentation:** 1 poor
**Contribution:** 2 fair
**Rating:** 3
**Confidence:** 4

**Summary:**

This paper proposes a curriculum transfer method, by decomposing a hard task into several simpler ones, learning on simpler tasks first and transferring knowledge to harder tasks. Experiments are conducted on classic Gym environments and one real-world control task.

**Strengths:**

The research direction of curriculum transfer learning is an important one that may inspire the community.

The environment considers one real-world control task.

**Weaknesses:**

However, the reviewer finds this paper hard to follow, and it requires significant improvements in clarity.

1)  A lot of technical details are missing or need to be clarified. For example, a) the paper mentioned that 'source tasks are sampled based on their complexity' many times. But what's the sampled policy? b) Also, the paper mentioned that 'the state space is different, then a function $s_{\mathcal{M}}(\phi) = h(s_{\mathcal{M}_t}(\phi))$ is needed'. What does $h(\cdot)$ define? How to learn $h(\cdot)$? Is it pre-defined by humans? What's $\phi$? c) In Section 4.2, does the equation update the source Q-functions? Why? d) The formula '$\alpha +\beta=1$ and $\alpha>\beta=1$ ' does not make sense. e) The way to combine a set of Q-functions into one source Q-function is not clear.

2)  The contents are inconsistent. For example, 'self-correction' and 'self-correcting' are mixed-used.
3)  Some typos exist in the paper, making it hard to understand.
4) The figures are hard to read.

No limitations are discussed in this paper. For example,
1) Using $\max(Q)$ makes the paper not able to extend to continuous action spaces.
2) The curriculum is defined by humans, which is infeasible to be applied to complex domains, requiring a certain amount of expert domain knowledge.
3) There are a lot of related works doing the same thing as this paper [1-3], while they automatically generate the curriculum.

[1] Emergent complexity and zero-shot transfer via unsupervised environment design - Dennis et al.NEURIPS 2020

[2] Evolving curricula with regret-based environment design - Parker-Holder et al. ICML 2022

[3] Understanding the complexity gains of single-task rl with a curriculum. ICML 2023.

**Questions:**

Please see the pros and cons part.

---

### Meta-Review · Area_Chair_A7wv · 2023-12-06

**Metareview:**

The paper proposes a new curriculum learning method where a target task is decomposed into a progression of simpler tasks by varying the degrees of freedom in the target task. The method operates by first learning simpler tasks and then transferring knowledge to harder tasks based on value functions. The reviewers acknowledged that the proposed idea of varying degrees of freedom and knowledge transfer for curriculum learning is interesting. However, the reviewers pointed out several weaknesses in the paper, and there was a consensus that the work is not yet ready for publication. The reviewers have provided detailed and constructive feedback to the authors. We hope the authors can incorporate this feedback when preparing future revisions of the paper.

**Justification For Why Not Higher Score:**

There was a consensus among the reviewers that the work is not yet ready for publication.

**Justification For Why Not Lower Score:**

N/A

---

### Decision · Program_Chairs · 2024-01-16

Reject